# Concurrent and Subsequent Co-Infections of *Clostridioides difficile* Colitis in the Era of Gut Microbiota and Expanding Treatment Options

**DOI:** 10.3390/microorganisms10071275

**Published:** 2022-06-23

**Authors:** Mattia Trunfio, Silvia Scabini, Walter Rugge, Stefano Bonora, Giovanni Di Perri, Andrea Calcagno

**Affiliations:** 1Unit of Infectious Diseases, Amedeo di Savoia Hospital, Department of Medical Sciences, University of Turin, 10149 Torino, Italy; walterrugge@gmail.com (W.R.); stefano.bonora@unito.it (S.B.); giovanni.diperri@unito.it (G.D.P.); andrea.calcagno@unito.it (A.C.); 2Unit of Infectious Diseases, AOU “Città della Salute e della Scienza”, Department of Medical Sciences, University of Turin, 10149 Torino, Italy; silviascabini88@gmail.com

**Keywords:** *Clostridioides difficile*, co-infection, blood-stream infection, *Enterobacteriaceae*, *Candida*, probiotics, *Cytomegalovirus*, gut microbiota, microbial translocation, fecal microbiota transplantation

## Abstract

We narratively reviewed the physiopathology, epidemiology, and management of co-infections in *Clostridioides difficile* colitis (CDI) by searching the following keywords in Embase, MedLine, and PubMed: “*Clostridium/Clostridioides difficile*”, “co-infection”, “blood-stream infection” (BSI), “fungemia”, “*Candida*”, “*Cytomegalovirus*”, “probiotics”, “microbial translocation” (MT). Bacterial BSIs (mainly by *Enterobacteriaceae* and *Enterococcus*) and fungemia (mainly by *Candida albicans*) may occur in up to 20% and 9% of CDI, increasing mortality and length of hospitalization. Up to 68% of the isolates are multi-drug-resistant bacteria. A pivotal role is played by gut dysbiosis, intestinal barrier leakage, and MT. Specific risk factors are represented by CDI-inducing broad-spectrum antibiotics, oral vancomycin use, and CDI severity. Probiotics administration (mainly *Saccharomyces* and *Lactobacillus*) during moderate/severe CDI may favor probiotics superinfection. Other co-infections (such as *Cytomegalovirus* or protozoa) can complicate limited and specific cases. There is mounting evidence that fidaxomicin, bezlotoxumab, and fecal microbiota transplantation can significantly reduce the rate of co-infections compared to historical therapies by interrupting the vicious circle between CDI, treatments, and MT. Bacterial BSIs and candidemia represent the most common co-infections in CDI. Physicians should be aware of this complication to promptly diagnose and treat it and enforce preventive strategies that include a more comprehensive consideration of newer treatment options.

## 1. Introduction

*Clostridioides difficile* (previously known as *Clostridium difficile,* Cdiff) is a spore-forming, obligate anaerobe, Gram-positive bacterium found in the intestinal tract of both humans and animals, from where its spores are shed in the environment and survive in variable and extreme conditions [1]. Cdiff is well recognized as one of the main causes of healthcare-associated (HA) diarrhea, and this “superbug” has recently emerged also as a less common cause of community-acquired diarrhea in younger individuals lacking traditional HA risk factors [2]. The European Centre for Disease Prevention and Control reported in 2016 a total of 7711 cases of Cdiff infection (CDI) from 556 hospitals in Europe, of which 5765 (74.6%) were HA infections. In the US, after an initial increase in Cdiff incidence, a reduction in HA-Cdiff cases was observed between 2011 and 2017 [3]. An important rate reduction in HA-CDI was also observed during the COVID-19 pandemic in parallel with the implementation of contact and droplet preventive measures [4]. However, in the last 20 years, community-associated CDI has been increasing, accounting for approximately half of all CDI cases in the US [5].

Differently from the status of an asymptomatic carrier, the clinical spectrum of CDI may range from mild diarrhea to severe and life-threatening fulminant colitis with well-recognized complications such as sepsis, toxic megacolon, and perforation. Transmural pancolitis that may require emergency segmental or total colectomy has been reported [1]. The overall mortality due to CDI ranges from 2 to 6%, though it is significantly higher in patients with acute renal failure, underlying inflammatory bowel diseases, or infections due to highly virulent Cdiff strains [6]. In fact, several strains have been associated with different entities of clinical severity. The Cdiff BI/NAP1/027 strain is known to be more virulent than others, which seems to be attributable to its increased toxins production [7]. The widespread use of fluoroquinolones has been strongly correlated with the emergence of this strain, and, after the 2000s, the number of cases due to Cdiff 027 has increased dramatically all over the world [8]. Another ‘hyper-virulent’ strain is the ribotype 078, which has been increasingly identified in community-acquired CDI and is genetically similar to swine isolates, suggesting a food-borne or zoonotic origin and a subsequent circulation in the community [9].

There are several well-known risk factors involved in the acquisition of Cdiff and in the development of CDI, including admission to healthcare facilities, older age, gastric acid inhibition/reduction, comorbidities such as inflammatory bowel disorders, and most notably, broad-spectrum antibiotic use. Exposure to clindamycin, fluoroquinolones, and beta-lactam/beta-lactamase inhibitor combinations has been associated with the highest risk as compared to macrolides, sulfonamides, and penicillin [6]. On the contrary, some pharmaco-epidemiologic studies have shown a protective effect from statins and probiotics, but this issue remains controversial [10,11]. Recent studies suggest that acid suppression with proton pump inhibitors is correlated with an increased incidence of Cdiff infection. While spores are resistant to acid, the vegetative form may survive in gastric contents with increased pH. The use of PPIs may also promote the expansion and colonization of Cdiff by its recognized potential to induce small bowel bacterial overgrowth with anaerobic colonic organisms [12]. Another less recognized risk factor for Cdiff is the prolonged use of elemental diets, such as enteral feeding, that act through alterations in gut microbiota composition. Such diets are totally absorbed within the small intestine and therefore deprive the colonic microbiota of their source of nutrition, such as dietary fiber, fructose oligosaccharides, and resistant starch, leading to suppression of colonic fermentation and synthesis of butyrate (a short chain fatty acid). Butyrate deficiency affects microbiota balance and promotes the creation of a “permissive” environment for Cdiff colonization [13]. All these risk factors underline how important is gut microbiota in allowing colonization and disease development for *Clostridioides* spp.

In view of the severity and mortality related to CDI, it is fundamental to be familiar with any factors that may further affect the burden of this severe cause of increased in-hospital mortality and prolonged hospitalization. A potential frightening complication of CDI is the development of concurrent or subsequent infections due to a large spectrum of microbial entities, including other bacteria, viruses, and fungi. The contribution of microbial translocation to co-infections occurrence in other infectious diseases such as *Strongyloides stercoralis* hyperinfection or severe Dengue has already been detailed [14,15]. Little is known about the mechanisms and frequency of concurrent and subsequent infections, as well as their impact on the overall clinical outcomes of Cdiff infection in terms of survival, complications, and recurrence. Current national and international guidelines do not have specific recommendations on how to assess nor prevent this event. This may be explained by the fact that the mechanisms underlying this phenomenon are still unclear, and a study to assess the causation between Cdiff infection and the occurrence of co-infections has never been carried out. Therefore, an organized description of this phenomenon has not been systemized yet, and scattered evidence is reported in literature. Plausibly, Cdiff per sè, as well as common risk factors of Cdiff colonization and reactivation, may predispose to microbial translocation (defined as the migration of bacteria, fungi, and/or their products from the gut lumen to extraintestinal space and systemic circulation), blood-stream infections (BSIs), or reactivation of other gut pathogens, as summarized in Figure 1. Highlighting common pathogenic pathways may be helpful in improving the clinical management of CDI by prompt recognition and management of such co-infections.

We searched for articles indexed in Embase, MedLine, and PubMed and published in English up to January 2022 by using the following keywords: “*Clostridium/Clostridioides difficile*”, “co-infection”, “blood-stream infection”, “fungemia”, “*Candida*”, “*Cytomegalovirus*”, “probiotics”, and “microbial translocation”. In this narrative review, we provided a summary of the existing evidence on concurrent and subsequent co-infections associated with CDI trying to describe the overall burden and mechanisms of this phenomenon, to assess whether co-infections have a relevant impact on the outcomes and management of patients suffering from CDI, and to provide some practical clinical considerations.

## 2. Bacterial Blood-Stream Infections and *Clostridioides difficile*

It is well established that the hematogenous translocation of bacteria residing in the gut is favored by some conditions: the loss of integrity of the intestinal mucosal barrier, the alterations of mucosal immunity, and the colonization of gut by overgrowing pathogens. During CDI, all these mechanisms take place along with an important mucosal inflammatory response [16]. Cdiff toxins A and B, which are primarily responsible for tissue damage and associated symptoms, stimulate inflammatory responses in the colonic lining by inducing cytoskeletal changes that compromise the epithelial barrier and stimulate inflammatory cytokines production. The disruption of enterocyte tight junctions allows toxins to cross the epithelium, where they can further induce immune responses in the cells residing in the lamina propria, leading to marked neutrophil recruitment and further destruction of the intestinal lining; indeed, the final pathological hallmark of CDI is the formation of pseudomembranes (Figure 1) [17]. 

While human data are scarce, murine models have already described a potential role of all these Cdiff-induced intestinal alterations in favoring microbial translocation and subsequent BSIs [18]. It has been shown that the development of CDI may be favored by perturbations in gut microbiota which in turn is further altered by CDI, starting a vicious circle (Figure 1). In this setting, *Bacteroidetes* and *Bifidobacterium* spp. play an important role in the mechanism of resistance to Cdiff colonization. Lower concentrations of *Bacteroidetes* and higher relative amounts of *Firmicutes* and *Proteobacteria* were found in the gut of patients with CDI compared to controls [19]. Similarly, few studies have shown how Cdiff can alter the composition of gut microbiota and promotes colonization with multidrug-resistant (MDR) organisms [20]. Furthermore, MDR organisms’ selection, gut dysbiosis, and eventually microbial translocation and BSIs can also be favored by some of the predisposing mechanisms leading to CDI, such as specific patterns of residing gut microbiota and antibiotics administration, amplifying the vicious circle that links CDI to BSIs (Figure 1). To further aggravate this event, even the use of anti-Cdiff therapies may contribute to alterations in the intestinal flora, to the overgrowth of bacterial populations, and, eventually, to the facilitation of bacterial translocation, microbe dissemination, and sepsis (Figure 1) [21].

Despite all these possible mechanisms involved in BSI development, the incidence and impact of BSIs complicating CDI have not been properly characterized to date. In Table 1, we have summarized the available studies from the literature that reported specific period prevalence of either or both bacterial and fungal blood-stream infections after CDI. 

Falcone et al. described for the first time a significant association between CDI and subsequent HA-BSI [23]. In this retrospective analysis of 393 cases, 18.3% developed HA-BSI within 30 days from the onset of CDI. BSIs were caused by enteric pathogens (*Candida* spp, *Enterobacteriaceae,* and *Enterococcus* spp.), and 68.4% of the microbial isolates were MDR. Thirty-day mortality was significantly higher in the CDI plus BSI group compared to the CDI-only group (38.9% vs. 13.1%), as well as the incidence of intensive care unit (ICU) admission and longer hospitalization length. Higher oral vancomycin dosage (>500 mg/day), infection by Cdiff ribotype 027, Cdiff recurrence, and severe colitis were found to be independent risk factors for HA-BSI [23]. 

Similarly, in a study investigating non-staphylococcal BSIs in relation to the time from the first Cdiff-positive fecal sample, bacteremia from unrecognized sources (occult BSI) occurred more frequently from 3 days before to 10 days after Cdiff toxin positivity compared to the pre-Cdiff period. Of note, during the Cdiff period, positive blood cultures were characterized by a greater percentage of enterococci (50%), and the majority of occult BSI resolved without treatment [25]. In line with these findings, CDI has been identified as a risk factor for vancomycin-resistant enterococci (VRE) bacteremia in a small cohort (*n* = 59) of patients with acute leukemia [27]. On the other side, it is not clear whether VRE gut colonization increases the risk of Cdiff colitis, or it can be favored by vancomycin-based treatments for CDI. To date, we can only observe that Cdiff co-colonization (and eventually co-infection) is more common in patients with VRE infection/colonization [28,29].

Conversely, another group observed only 86 cases of BSIs in a cohort of 570 patients with CDI (7.6%). *Enterococcus* and *Klebsiella* spp. were the most common bacterial isolates (14% for both) [26]. Patients with BSIs showed a higher prevalence of comorbidities, and they were more likely to be immunosuppressed, critically ill, and to have a central venous catheter (CVC) in place. Surprisingly, CDI appeared protective against subsequent BSIs at the multivariate model after adjusting for gender, Charlson Comorbidity score, systemic inflammatory response syndrome, and CVC. The authors hypothesized that systemic immune activation and inflammation triggered by colitis itself could favor the clearance of blood-stream pathogens. However, although the large sample, this study was limited by the retrospective design, the length of follow up limited by the average hospital stay, and the possibility of very complex multifactorial confounders, since they did not rule out infectious sources other than the gastrointestinal tract despite the presence of staphylococcal BSI and CVC [26]. In opposition to this hypothesis, Oliva et al. found that among 45 subjects hospitalized for CDI, of whom 17.7% developed BSIs, markers of microbial translocation, inflammation, and intestinal damage were increased during CDI, decreased after treatment, and did not normalize compared to healthy controls after CDI resolution [22]. Subjects developing BSIs had higher microbial translocation and maintained it at higher degree after CDI resolution compared to those not complicating with BSIs, suggesting that local and systemic inflammation associate with intestinal barrier disruption, microbial translocation, and eventually increased risk of co-infections.

Cdiff bacteremia is also possible [30], and in the case of a prominent intestinal barrier injury as the main underlying mechanism for BSI co-occurrence, it should be expected at a similar rate to BSIs. Nevertheless, in the literature, cases of BSIs directly due to Cdiff isolated in blood cultures are extremely rare and solely reported in patients with underlying relevant gastrointestinal disorders. In these cases, BSIs are mixed bacterial co-infections with Cdiff and other gut bacteria, suggesting that massive intestinal barrier dysfunction is required for the translocation of Cdiff. Indeed, it is also possible that being Cdiff an anaerobe, some BSIs diagnoses may miss this blood co-infection due to a relatively higher difficulty in cultivating this germ [31]. 

Mortality in CDI is likely multifactorial. It is possible that concurrent bacterial translocation contributes to this, but it is hard to precisely estimate its effect, considering that BSIs seem to mainly complicate the more severe CDI only. Considering that both CDI infection and subsequent BSIs represent complex interactions between pathogens, host, native microbiota, and its perturbations and that most risk factors for CDI and BSI due to gut translocation overlap, further studies are needed to investigate this relationship and plan strategies to prevent BSIs in high-risk patients. In the latest guidelines on the management of Cdiff in adults released by the Infectious Diseases Society of America (IDSA) and the Society for Healthcare Epidemiology of America (SHEA), as well as in the latest from the European Society of Clinical Microbiology and Infectious Diseases (ESCMID), fidaxomicin is suggested as the preferred agent for initial CDI and for the first recurrent episode [32,33]. Fidaxomicin significantly reduces the recurrence rate in most patients compared to vancomycin, while it is non-inferior in terms of clinical cure [33]. This update in recommendations could also embrace the call to reduce the risk of concurrent and subsequent BSIs, thanks to the minimal impact of fidaxomicin, an oral macrolide, on gut microbiota compared to metronidazole and oral vancomycin. While the choice of fidaxomicin as first-line therapy for CDI is still limited among physicians partially due to its higher cost, the potential shorter length of hospitalization and reduced incidence of complicating BSIs (that have still to be demonstrated) may be cost-effective.

## 3. *Candida* spp. and *Clostridioides difficile*

Candidemia is defined as the presence of *Candida* spp. in blood, and it is invariably a pathological condition that requires proper evaluation, prompt treatment, and risk factor management as it cannot ever be considered a simple contamination. 

*Candida* spp. is part of the normal gut microbiota, and translocation through the gastrointestinal wall is probably the most common mechanism by which *Candida* spp. enters the blood-stream in fragile patients, such as neutropenic subjects or those admitted to ICUs. The physiopathology behind the occurrence of candidemia in CDI should be the same as the one underlying bacterial BSIs. Raponi et al. have highlighted how CDI can predispose to *Candida* spp. overgrowth and its subsequent spread into the blood. In this prospective case-control study, they found that CDI was significantly associated with gut colonization by *Candida* spp. (83% in CDI-positive vs. 67% in CDI-negative), with *Candida albicans* being the species most often implicated [34]. Once again, the more plausible reasons for this phenomenon can be attributed to antibiotics use against both Cdiff and/or the concomitant infections that precede Cdiff colitis by reducing gut commensal competitors, as well as to direct interactions between *Candida* spp. and Cdiff. As for CDI-associated antibiotics, Nerandzic et al. analyzed the differences in the alterations of intestinal flora following different antibiotic treatments for CDI. After fidaxomicin treatment, there was a significant reduction in the risk of colonization by VRE or in the overgrowth of *Candida* spp. compared to patients receiving oral vancomycin; a finding likely attributable to the different spectrum of activity on the intestinal anaerobic flora [35]. On the other side, the immunological alterations and changes in gut microbiota induced by *Candida* spp. overgrowth can modulate the susceptibility to CDI. In an experimental animal study, after oral inoculation of Cdiff spores, a lower rate of death due to CDI was observed in infected mice pre-colonized with *C. albicans* compared to those without. The subsequent growth of Cdiff in the gastrointestinal tract, the production of its toxins, and the presence of inflammation and tissue damage were similar in both groups, but the expression of specific inflammatory cytokines (such as IL-17A) in the infected tissues differed (being higher in pre-colonized mice, suggesting a different host response to Cdiff according to the presence and amount of *Candida* spp.) [36]. On the contrary, another mouse model detected higher serum 1–3 β-D-glucan (BDG), a fungal cell wall component, spontaneous Gram-negative BSIs, and gastrointestinal leakage markers in Cdiff infected mice that died compared to those surviving [37]. In this model, BDG resulted as the best prognostic biomarker for 7-day mortality, and its levels, along with CDI severity, were attenuated by pre-emptive treatment with *Lactobacillus rhamnosus* [37]. In another experimental study on the physical and chemical interactions between Cdiff and *C. albicans*, Cdiff was able to thrive at ambient oxygen levels when co-cultured with *C. albicans*, and it could secrete a compound with inhibitory activity against two virulence factors of *C. albicans* that modulate the transition from yeast (the invasive form) to hyphae and biofilm formation [38]. Therefore, further studies are required to clarify what type of interactions between Cdiff and *Candida* spp. may occur and which are the resulting consequences to the host, as apparently discordant preliminary data point towards enhanced severity of the clinical co-infectious episode, but also to a concurrent increase in *Candida* spp. invasiveness and reduced virulence in the presence of actively replicating Cdiff.

Overall, *Candida* spp. seems to be the single most common microbial isolate in blood during CDI, and the prevalence of candidemia following CDI varied considerably among studies (0.8–8.6%), but co-infection is associated with substantially increased mortality [24]. Candidemia-related mortality is approximately 40%, but when candidemia is secondary to CDI, mortality can reach up to 60% [39]. This may be due to a higher translocating microbial burden, increased mucosal injury, an exacerbated inflammatory gut milieu, and the fact that candidemia is more commonly found in severe colitis among CDI cases. Indeed, severe CDI (aOR 4.4), including those by 027 ribotype (aOR 4.5), relapsing CDI (aOR 5.9), the treatment with high doses of vancomycin (≥1000 mg/day, aOR 2.1), immunosuppressive therapy (aOR 2.2), and the number of CDI relapses (aOR 3.1) have been previously recognized as independent risk factors for candidemia [40]. Similarly, treatment with vancomycin plus metronidazole (usually prescribed for more severe cases) and severe CDI (based on clinical evaluation and Cdiff-specific complications) were associated with higher odds of developing candidemia up to 120 days after a Cdiff episode [24]. Therefore, we believe that a high index of suspicion for invasive candidiasis should be recommended, especially after a severe CDI episode.

## 4. *Cytomegalovirus* and *Clostridioides difficile*

*Cytomegalovirus* (CMV) infection is an important cause of morbidity and mortality among immunocompromised patients in contexts such as organ transplantation, chemotherapy, inflammatory bowel diseases (IBD) receiving immunosuppressive agents, and HIV infection [41]. Gastrointestinal involvement by CMV, namely CMV colitis, remains a rare occurrence in an immunocompetent host, but it is increasingly recognized in apparently immunocompetent subjects with some immune-modulating conditions such as advanced age, chronic renal failure, diabetes mellitus, and prolonged ICU stay [42].

Like the clinical manifestations of CDI, CMV colitis can manifest with symptoms such as diarrhea, abdominal pain, weight loss, intestinal bleeding, or fever and lead to complications such as toxic megacolon or even bowel perforation. Despite the aforementioned non-traditional risk factors for CMV colitis (partially overlapping with those of CDI) very few cases of this co-infection have been reported to date, so the incidence of this phenomenon cannot be inferred. Nevertheless, in the context of IBD, a close relationship between disease flares and CMV replication has been documented as gut inflammation seems to favor herpetic reactivation from compartmentalized intestinal sites of latent infection [43]. The mechanisms underlying potential reactivation of intestinal latently residing CMV during Cdiff infections are not clear, but the shift in the mucosal immunologic balance, both in terms of cells and of cytokine patterns, could be hypothesized as it has been initially described in IBD and lead us to include CMV among the co-infections that should be considered. Moreover, it is possible that this co-infection is underestimated since the diagnosis of CMV colitis can be easily missed in patients with severe diarrhea and positive Cdiff toxin without apparent immunologic conditions requiring further investigations. In line with this hypothesis, the co-infection has been mainly described in case reports on CDI refractory to proper treatments that underwent further diagnostic work-up. In a patient with a squamous cell carcinoma of the lip and pancolitis secondary to CDI, persisting diarrhea despite appropriate treatment and proven bacteriological cure for Cdiff led to testing the stool and blood for CMV-DNA, which resulted in positive and dramatically improved after ganciclovir administration [44]. Florescu et al. reported two cases of Cdiff and CMV co-infection in solid organ transplant recipients and analyzed seven previously published reports. Surprisingly, the authors observed a 100% rate of positive blood PCR for CMV, raising the possibility that CDI may increase the chance of developing a detectable viremia [45]. Unfortunately, no values of the detected viremia were reported, so it cannot be assumed a real pathogenic role of plasma viremia; indeed, the role of low CMV viremia is uncertain in immunocompetent subjects or in HIV-positive patients with no evidence of organ involvement by CMV [46,47].

Few cases of CMV colitis following successful therapy for CDI have also been described: one patient admitted to ICU had colic ulceration due to CMV colitis three weeks after the resolution of a properly treated CDI, while a second elderly case showed CMV colitis as the cause of persisting bloody diarrhea at the end of oral vancomycin treatment for CDI [48]. 

Although the co-infection of Cdiff and CMV seems to be a rare entity, considering the synergistic activity of these pathogens in increasing the risk of lethal intestinal perforation and toxic megacolon, clinicians should have a high index of suspicion to rule out CMV reactivation even in immunocompetent hosts, especially when other comorbidities or refractory disease are observed. 

## 5. Other Co-Infections in *Clostridioides difficile* Colitis

Co-infections with other enteric pathogens such as *Salmonella* spp.*, Cryptosporidium* spp.*, Giardia* spp.*, Enterocytozoon* spp., and *Campylobacter* spp. have been described both in community and hospital-acquired cases of CDI [49,50]. In a prospective study on adult patients in Scotland, 13.3% of the tested HA-CDI cases were found to be co-infected with norovirus, which is a common pathogen causing nosocomial outbreaks [49].

Very few data are available about the co-infection of Cdiff with other common intestinal parasites. For instance, *Entamoeba histolytica* and Cdiff may present with similar clinical features or endoscopic findings and considering that empiric use of metronidazole for colitis treatment is widely practiced in low- and middle-income countries without testing for infectious causes, it may be speculated that amoebiasis is over-diagnosed and Cdiff infection underestimated as the latter is seldom considered and probably treated unknowingly with metronidazole [51]. Existing data suggest that the burden of CDI in low/middle-income countries is similar to high-income countries, but in the former, the diagnosis is hampered by both the lack of available testing and a low index of clinical suspicion [52]. Considering that the prevalence of intestinal parasites is higher in this setting, co-infections with Cdiff may not be an infrequent occurrence.

Co-infection with Cdiff and other gastrointestinal pathogens may also be common in children suffering from diarrhea, with a reported pooled rate of co-infections of 20.7% in Cdiff-positive children [53]. Viral co-infections seem to be the most commonly found (46.0%), while bacterial and parasitic co-infections accounted for 14.9% and 0.01% of the cases, respectively [53]. Unfortunately, the included studies were not conclusive regarding the impact of co-infections on CDI severity, and none evaluated causal relationships. Very scarce data also exist on these co-infections in adults. A case of co-infection with *Giardia lamblia* and Cdiff was described in a 49-year-old man taking ranitidine [54]. In another case, a patient with colorectal cancer was reported to have co-existing Cdiff and intestinal amebiasis infection [55]. Due to the limited data available to date, no evidence can suggest an increased incidence or severity of parasitic intestinal infections in the presence of Cdiff/CDI.

## 6. Probiotics and *Clostridioides difficile*

Probiotics may play a role in the prevention of CDI by several mechanisms, including colonization resistance through maintaining a healthy gut flora, enhancing the clearance of Cdiff at the end of treatment, and inactivating the toxin receptor sites before the germination and growth of spores in the colon [56]. Nevertheless, the use of probiotics in routine clinical practice remains debated. 

A Cochrane meta-analysis of 39 randomized clinical trials concluded that probiotics reduce the incidence of CDI by 70% in adult and pediatric patients undergoing antibiotics for any reason, providing moderate quality of evidence in support of probiotics use in preventing Cdiff colitis. Post hoc analysis indicated that probiotics actually show preventive efficacy in patients with at least mild-moderate baseline risk of CDI and no benefit for subjects characterized by low risk [57]. Conversely, a recent retrospective study on more than 3000 adults hospitalized patients observed that patients who received antibiotics with concurrent administration of probiotics (mainly *Lactobacillus* spp.) were more likely to develop CDI compared with those who did not receive probiotics (HR 2.7) [58]. Similarly, the use of probiotics was not associated with decreased incidence of CDI among hospitalized adults aged 50 and above who received antibiotics in a recent multicentric study [59].

It is also recognized that these preparations, containing living microorganisms, can uncommonly cause different forms of invasive infections, particularly in critically ill or severely immunocompromised patients. Cases of *Saccharomyces cerevisiae* fungemia have been reported in patients with a history of probiotic use [60]. Furthermore, there is evidence of potential nosocomial development of fungemia in wards where probiotics were used: the contamination at the sites of vascular access was identified as the probable mechanism by which probiotics caused BSIs [61]. As for other bacterial and fungal BSIs, also intestinal barrier impairment and concomitant administrations of broad-spectrum antibiotics have been acknowledged as possible risk factors for probiotics bacteremia, raising the hypothesis of an intestinal source that may play a role even in CDI [62]. In line with this, some reports suggest the possibility of developing *S. cerevisiae* or *Lactobacillus rhamnosus* fungemia/bacteremia when probiotics containing such microorganisms were administered during CDI. In the latter case, bacteria could have been selected by the prolonged oral vancomycin therapy the patient received along with live yogurt as the administered probiotic (since *Lactobacillus* spp. are intrinsically resistant to vancomycin) [63]. Accordingly, cases of sepsis due to *Lactobacillus* spp. have been reported in neutropenic patients after oral vancomycin [64].

While probiotics may play a role in preventing CDI in patients at risk that have not yet developed the infection, the administration of probiotics during overt and ongoing Cdiff colitis is therefore controversial and may be risky. Since the combination of enhanced intestinal permeability, altered gut microbiota, and immunosuppression is present in a large proportion of patients affected by CDI, the use of probiotics in this setting should be further evaluated to properly balance risks and benefits, especially when an extended duration of vancomycin is administered.

Finally, after a severe episode of CDI, immunological perturbations and cell damage in the intestinal mucosal barrier can occur and persist for several weeks [22]. This functional disbalance in the gut and systemic immunity, already described in many other severe infections (such as malaria and septic shock), can also cast shadows on the opportunity and safety of probiotics to recover the normal gut microbiome following a severe episode of CDI. In a recent phase 3, double-blind RCT, the oral administration of SER-109, an investigational microbiome therapeutics composed of purified *Firmicutes* spores, to patients healed from a third or further episode of CDI (after standard-of-care antibiotic treatment) reduced the relative risk of recurrent infection by about 70% compared to placebo [65]. The study population was represented by 99% of outpatient subjects; therefore, it is likely that these promising results can be reliably applied to non-severe recurrent CDI (rCDI), and future studies are required to assess the safety and usefulness of probiotics in post-severe CDI.

## 7. Fecal Microbiota Transplantation in *Clostridioides difficile* Colitis

Fecal microbiota transplantation (FMT), defined as the transfer of fecal microorganisms from healthy donors into the gut of recipient patients, has been associated with robust efficacy in the treatment of rCDI [33]. One or two FMT can be sufficient to cure rCDI in 90% of cases [66]. In a randomized clinical study including 64 adult patients with rCDI, FMT delivered by colonoscopy or naso-jejunal tube after a short course of vancomycin was superior to fidaxomicin and standard-dose vancomycin monotherapies, based on the endpoints of clinical and microbiological resolution or clinical resolution alone [67].

Recent data suggest that FMT may be an alternative to antibiotic therapy also in the first CDI episode. In a small trial investigating the efficacy of FMT as a treatment for primary CDI, a clinical cure after initial treatment with no evidence of recurrence was achieved in seven patients in the transplantation group (78%) as compared with five in the metronidazole group (45%) [68]. Additionally, a phase three trial to assess FMT as a first-line treatment for severe primary CDI is ongoing (NCT02301000). 

Interestingly, in a nonrandomized prospective single-center study, compared to antibiotics use, FMT reduced by 23% the incidence of BSIs in rCDI [69]. A higher proportion of patients had sustained cure of CDI after treatment in the FMT group than in the antibiotic group (97% vs. 38%); no patient in the FMT group required surgery for severe CDI compared with 14 subjects in the antibiotic group (0% vs. 8%), and the incidence of BSIs during the 90-day follow-up was lower in the FMT group compared to antibiotic group (5% vs. 22%, and 1% vs. 6% polymicrobial infections) [69]. No patient in the FMT developed fungal BSIs, while 12 (7.0%) cases of fungemia occurred in the antibiotic group [69]. Moreover, patients in the FMT group had a significantly shorter length of hospitalization (13.4 vs. 27.8 days) [69]. These results can be explained by several potential differences between the mechanisms underlying FMT compared to antibiotics: the earlier restoration of healthy gut microbiota, the avoidance of vancomycin, the increase in gut commensal competition, and the decrease in gut resistome, intended as the expression of antibiotic resistance genes by the gut microbiota. Similarly, despite the small sample (45 patients) and the tinier number of subjects receiving FMT for CDI (6 patients), compared to other treatments (vancomycin, fidaxomicin, and metronidazole), none of the subjects undergoing FMT developed BSIs [22].

Another possible favorable mechanism of prevention of co-infection operated by FMT is represented by the modulation of fecal bile acid composition; secondary bile acids, which are products of microbial metabolism, have been shown to inhibit Cdiff germination, growth, and toxin activity. The loss of beneficial *Firmicutes* bacteria induced by antibiotics leads to increased primary bile-acid concentrations, which on the contrary, enables Cdiff spore germination [70]. 

Further studies are warranted to assess the potential multiple properties and mechanisms by which FMT may affect intestinal damage, inflammation, microbial translocation phenomena, and eventually, co-infection rates compared to standard of care for primary and rCDI episodes.

## 8. Practical Considerations 

Based on currently available evidence, as well as on the relevant gap of data, it should be important to draw physicians’ attention to the possibility of concurrent and subsequent co-infections in CDI, as bacterial BSIs and Candidemia may occur in up to 20% and 9% of the cases, respectively (Figure 2).

Before, during, or up to one month after Cdiff treatment, fungal and bacterial co-infections should be suspected in the event of any clinical change or an unexpected variation in blood tests. As an example, high fever is an infrequent sign in CDI; it should trigger an assessment to rule out common co-infections, especially in non-severe cases (or whit deteriorating clinical status). Fulminant colitis is a life-threatening complication of CDI, occurring in about 3% of patients, and it is characterized by a clinical picture resembling that of septic shock: hypotension with or without the use of vasopressors, ileus, toxic megacolon, mental status changes, serum lactate levels >2.2 mmol/l, or any evidence of end-stage organ failure [71]. Of note, a substantial number of patients with fulminant CDI (36% to 75%) have a history of recent surgery [72], which is a common risk factor for invasive candidiasis. All these three diagnoses (fulminant colitis, MDR-BSI, and candidemia) should be considered and potentially empirically addressed in case of shock development. Similarly, the mean duration of CDI symptoms is variable, also depending on the antibiotic therapy, and early surgical consultation is recommended in patients who do not respond to conventional therapy within 3 days [73]. The median length of Cdiff-related diarrhea was reported to be shorter in patients treated with vancomycin (about 3 days) compared with those given metronidazole (about 5 days) [74]. The persistence or the new onset of signs and symptoms after 3–5 days from treatment initiation should also prompt the physician to rule out co-infections.

Clinical worsening after the resolution of a CDI episode should be interpreted as a warning sign to consider other explanations than Cdiff recurrence. Up to 25% of patients can experience recurrence of CDI within 30 days of completing treatment when antibiotic-induced microbiota disruption facilitates Cdiff spore germination, especially in the elderly, or persisting use of antibiotics and proton pump inhibitors after the diagnosis. After the second CDI episode, the risk of multiple recurrences increases to 40–65% [75]. In the event of signs or symptoms of infection or of relapsing diarrhea, the differential diagnosis should address both subsequent co-infections and Cdiff recurrence. In case of worsening or recurrent diarrhea during or after treatment, detection of CMV-DNA by real-time PCR in fecal and blood samples or microbiological investigations in stools may be worthy in selected cases presenting traditional and non-traditional risk factors for CMV colitis or for rarer gastrointestinal pathogens (see Figure 2 and Figure 3).

Prolonged diarrhea (defined as >5 days after the beginning of proper anti-Cdiff therapy) or relapsing diarrhea after initial resolution should also require the evaluation for causes other than recurrency and co-infections. After ruling out these events, in the absence of alternative diagnoses, patients suffering from persisting diarrhea may have refractory CDI or inflammatory colitis such as post-infectious irritable bowel syndrome; while the latter may complicate CDI in 4–25% of cases [71], it is extremely hard to exactly quantify the incidence and prevalence of refractory CDI as in most of the studies the temporal detection and definition criteria limit the possibility to distinguish it from reinfection. Detecting Cdiff toxins in the stool of these patients may not always be informative on the real ongoing pathological process, and colonoscopy should be considered. A practical scheme for the management of CDI and co-infections is depicted in Figure 3.

Together with clinical monitoring, a few common blood tests may help in assessing co-infections risk. Procalcitonin (PCT) remains at relatively low levels in CDI, although PCT concentration >0.5 μg/mL has been proposed as a reliable marker to identify severe CDI [76]. Monitoring PCT in addition to blood culture collection from febrile patients may be helpful since a significant PCT elevation can predict the presence of BSIs after the start of anti-Cdiff therapy. Furthermore, Gram-BSIs have significantly higher PCT concentrations than Gram+ BSIs and candidemia, allowing for potential microbial etiology stratification [77,78]. Strict monitoring of BDG levels can also be useful to rule out candidemia in emergent or persistent fever after initiation of anti-Cdiff therapy, especially in the setting of negative blood cultures. Lastly, marked leukocytosis can be seen in both candidemia and CDI, while leucopenia is more common in Gram-septicemia. 

CDI is an extreme ominous example of negative pharmacoenosis [79], where the potentiality of beneficial drug combos is still unsatisfactorily exploited both as treatment and prevention. In the event of BSIs, the need to use additional antibiotics (other than those for CDI) can increase the risk of prolonged diarrhea and CDI recurrence. Thus, antibiotics associated with a lower risk of CDI should be preferred (such as macrolides, aminoglycosides, sulfonamides, vancomycin, or tetracyclines) and discontinued as soon as possible. Some authors have suggested prolonging treatment with antibiotics acting against Cdiff for a week after another broad-spectrum therapy has been withdrawn [80]. Notably, differently from what occurred with other cephalosporins, ceftobiprole was shown to have no significant ecological impact on the human microbiota and to exhibit some inhibitory activity against Cdiff in experimental models [81,82]. Despite the role of tigecycline in CDI is controversial and not recommended by current guidelines [83], it may be considered as a potential therapeutic option for patients with severe CDI, in addition to standard therapies, and as a relatively safe treatment for sensitive bacterial co-infections. As an anti-CDI booster, we may consider the administration of high-dose tigecycline (200 mg loading dose, followed by 100 mg every 12 h) in severe and fulminant colitis, as recommended in critically ill patients with severe infections due to multidrug-resistant organisms [84]. 

Potential successful strategies to reduce the risk of co-infections should primarily rely on the use of targeted anti-Cdiff therapies, which have minimal impact on gut microbiota, such as fidaxomicin, bezlotoxumab, and FMT. This last approach might not be just considered as the salvage treatment following repeated failures of antibiotics but as a reliable therapeutic option, although adoption of FMT as a first-line treatment is not yet recommended, and it would require further clinical and cost-effectiveness assessments; indeed, promising but limited evidence on hypothesized relevant benefits of FMT over microbial translocation and risk of BSIs in CDI is there [22,85].

As for bezlotoxumab, none of the registration trials nor the few post-marketing studies reported on the incidence of BSIs in the study arms [86,87]. By blocking Cdiff toxin B, it is expected to act also dampening the gut epithelial injury and thereby the mechanical integrity of the mucosal barrier, reducing microbial translocation. On the opposite, vancomycin can be entero-toxic at high oral doses and inevitably increases gut dysbiosis and the risk of VRE colonization. Unfortunately, to date, oral vancomycin and bezlotoxumab do not occur at the same level in the management cascade of CDI, and further studies are required to endorse any reconsideration for the licensed prescription criteria of this anti-toxin B monoclonal antibody.

Prophylaxis with oral nystatin has been proposed in severe CDI at high risk for candidemia, but its role in preventing *Candida* translocation during and after Cdiff colitis has not been clearly established yet [39]. Probiotics seem beneficial in patients receiving antibiotic therapy at high risk for CDI, but evidence supporting their use as adjuvant therapy during overt CDI is scarce, and it is the opinion of the authors that during CDI, they may be potentially deleterious by increasing the risk of probiotics gut translocation and superinfections of blood-line access. Despite incomplete evidence, probiotics administration after the resolution of the episode may once again be promising in preventing at least CDI recurrences. Lastly, very few recent data point towards a potential application of prebiotics (either alone or in combination with probiotics) in preventing germination of Cdiff spores [88], modulating Cdiff adhesion [89,90], and in stimulating competing gut commensals [91] in both in vitro and animal models; thereby, prebiotics may indeed reduce CDI burden, but unfortunately to date, there are no studies reporting data on plausible effects of prebiotics on the modulation of the risk of microbial translocation and co-infections. 

## 9. Conclusions

In conclusion, further efforts are needed to better detail, quantify, prevent, and effectively manage the risk of concurrent and subsequent co-infections, mainly BSIs, in patients with CDI. Although it is not properly quantifiable to date, this complication likely contributes to the global burden of CDI by increasing the length of hospital stay and CDI-related mortality. Gut microbiota represents the main source of BSIs in CDI and its preservation, and eventually, restoration is pivotal to facing this clinical complication as well as to preventing Cdiff recurrences. Promising data on post-CDI-specific probiotics and FMT bodes well for the future. In the era of MDR bacteria, but also of expanding knowledge on gut microbiota and evolving treatment options, a multi-step approach to CDI and co-infections management is warranted; this should include CDI prevention by antimicrobial stewardship and modulation of risk factors, the adoption of microbiota “conciliating” drugs, the prompt recognition of this underestimated co-occurrence, and proper treatments able to avoid or limit the vicious circle that arises between Cdiff, antibiotics and gut environment.

## Figures and Tables

**Figure 1 microorganisms-10-01275-f001:**
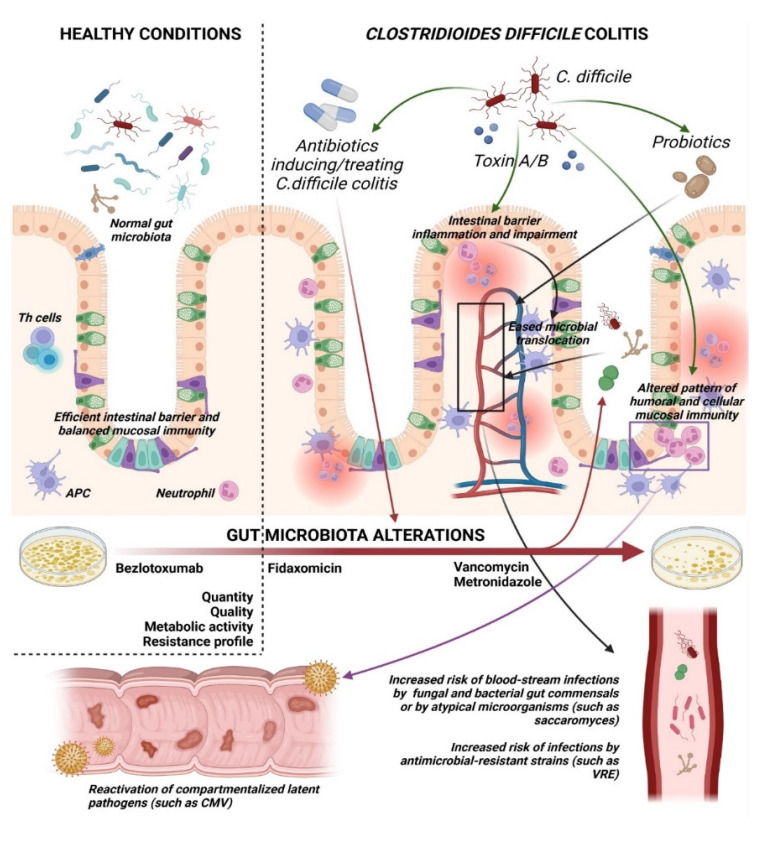
Schematic representation of predisposing factors and mechanisms underlying bacterial, fungal, and viral infections that develop concomitantly or subsequently to *Clostridioides difficile* colitis. *Legend: Th cells, T helper lymphocytes; APC, antigen presenting cells; CMV, Cytomegalovirus; VRE, Vancomycin-resistant Enterococcus* spp.

**Figure 2 microorganisms-10-01275-f002:**
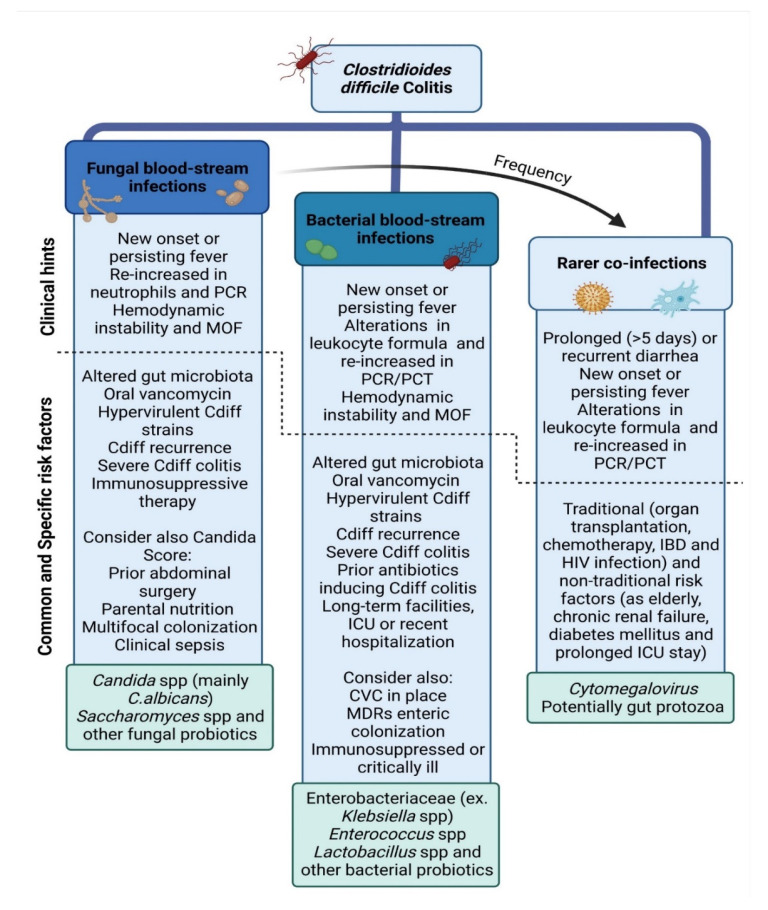
**Clinical changes, microbial epidemiology, and risk factors of co-infections in Clostridioides difficile colitis.** Legend: PCR, C reactive protein; MOF, multi organ failure; Cdiff, *Clostridioides difficile*; spp., species; PCT, procalcitonin; ICU, intensive care unit; CVC, central venous catheter; MDRs, multi-drug resistant bacteria; IBD, inflammatory bowel diseases; HIV, Human immunodeficiency virus.

**Figure 3 microorganisms-10-01275-f003:**
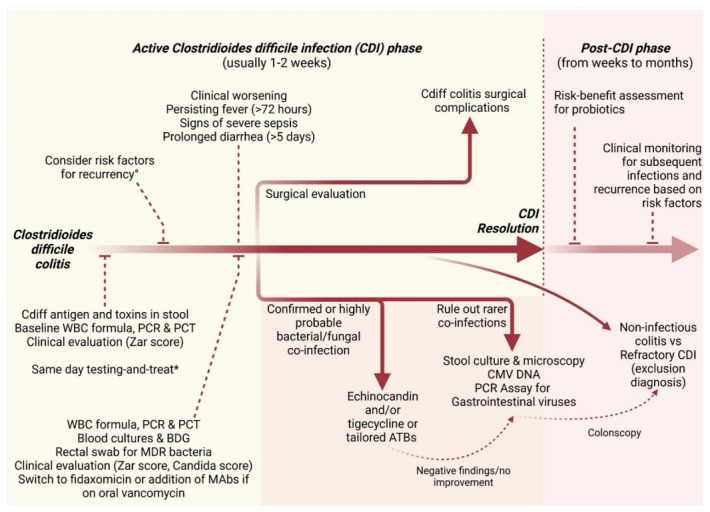
**Practical flow-chart on *Clostridioides difficile* and co-infections clinical management.** Legend: WBC, white blood cells; PCR, C reactive protein; PCT, procalcitonin; BDG, beta-D-glucan; MDR, multi-drug resistant; MAbs, anti-toxin B monoclonal antibodies (bezlotoxumab). * If possible prefer “microbiota-preserving” treatments, such as fidaxomicin.

**Table 1 microorganisms-10-01275-t001:** Summary of literature studies reporting period prevalence data on bacterial and/or fungal blood-stream infection during and following *Clostridioides difficile* colitis.

Work	N	Age * (Years)	Setting	BSI Prevalence and Isolates	From CDI to BSI (Days)	CDI Therapy	OP from CDI (Days)
[22]	45	75	Hospitalized (4% ICU)	**Overall**: 17.7% **bBSIs**: 15.6% (*K.pneumoniae*, *A.baumannii*, *E. faecalis* > *E. coli)* **Candidemia**: 6.7% (*Candida* spp.)	20.5 (9.7–35.7)	OV 93.3%, MET + OV 17.7% FMT 13.3% FDX 13.3%	60
[23]	393	74	Hospitalized (18% ICU; 19% surgery)	**Overall**: 18.3% **bBSIs**: 6.1% (*Enterobacteriaceae* > *Enterococcus* spp.) **Candidemia**: 8.6% (*C.albicans > C.glabrata* > other *Candida* spp.) **Mixed BSIs**: 3.6% (*Candida* spp.*, Enterococcus* spp., *K.pneumoniae* in different combinations)	NA	OV 82% MET 16% OV + MET (escalation) 32%	30
[24]	13,615	62	Hospitalized	**Candidemia**: 0.8% (37.2% *C.albicans*)	19 (8–45)	MET 40.7% MET + OV 39.8% OV 10.6%	120
[25]	505	NA	Hospitalized	**Overall**: 5.9% **bBSIs**: 2.9% (*Enterococcus* spp. *> Enterobacteriaceae > Corynebacterium, Pseudomonas*, *Clostridium, Streptococcus* spp.*, Lactobacillus* spp.*, Acinetobacter* spp.*, Pasteurella*) **Candidemia**: 0.6% (*Candida* spp.) **Mixed BSIs**: 2.4%	NA	NA	10
[26]	570	55	Hospitalized (29% ICU)	**Overall**: 6.3% Isolates NA	NA	NA	30

* Mean/median value. Legend: ICU, intensive care unit; BSI, blood-stream infection; bBSI, bacterial blood-stream infection; spp., species; CDI, *Clostridioides difficile* infection; OV, oral vancomycin; FMT, fecal microbiota transplantation; MET, endovenous metronidazole; FDX, fidaxomicin; OP, observation period; NA, not available.

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
