# Peer review of "Concurrent and Subsequent Co-Infections of *Clostridioides difficile* Colitis in the Era of Gut Microbiota and Expanding Treatment Options"

_microorganisms, 2022, doi:10.3390/microorganisms10071275_

Round 1

Reviewer 1 Report

There are many old references. Som of them need to deleted. 

Author Response

Thanks to Reviewer 1 for his/her time spent at improving our manuscript.

We do agree with the reviewer about the fact that only one third of our references belogns to publication period from 2018 to nowadays, nevertheless this is a narrative review without a time frame for studies inclusion and several specific data may have not been replicated since long time (murine models or prevalence studies as ex.). We believe that the reviewer 1's observation clearly underlies how scattered (even in time) is the available data and how poorly recognized and addressed are the issues discussed by this review. If the reviewer has some reference to suggest we are glad to include them in our manuscript. We did our best to include all the clinical studies reporting data on co-infections in CDI, whereas for case reports and animal/in vitro models we referenced those that best fitted for arguing or confirming clinical evidence. In this latter case the choice of references is more arbitrary and few papers could have been picken up from a period before 2000s, but this should not be a criteria for judging the quality of a reference (as example, we cited two papers published in 1997, but both are two case series on VRE bactermia-CID in leukemic patients and Lactobacillus bacteremia in neutropenic patients- so that newer publication on the same topic would not add much more than what has been described before).

While reviewing references we corrected a duplicate (IDS guidelines 2021 on Cdiff management are now only ref. 33 - before it was 32 and 33, apologies).

Thanks to Reviewer 1 once again for his/her time spent on our manuscript.

Reviewer 2 Report

In this study, Trunfio et al. reviewed the co-infections of C.difficile infection from multi-kingdom. I only have a few comments:

1. A table is needed to summarize the current study on co-infections of C.difficile infection, including sample size, ethnicity, sequencing platform, bacteria/fungi/virus, key findings, etc.

2. Line 222: "Candida spp and Clostridioides difficile", are there other fungi associated with CDI?

3. Authors should talk a little about which one comes first, C. difficile? or other pathogens?

4. Authors should also talk more about the intervention approach for CDI, such as antibiotics and prebiotics.

Author Response

  1. A table is needed to summarize the current study on co-infections of C.difficile infection, including sample size, ethnicity, sequencing platform, bacteria/fungi/virus, key findings, etc.

AR: Thanks to Rev.2 for her/his time spent at improving our work. We were in doubt since the first submission on whether to include or not such a Table. The main issue herein relates to the vast heterogeneity in the studies reporting co-infections “rates”; this would lead to include studies extremely different in study design so that a summary of the data would lead to wrong interpretations or odds difference. Indeed, since we do agree with Rev.2 that such an effort can improve the work, we have now include Table 1 but specifically reporting only studies that validly reported on period prevalence of co-infections in Cdiff (we excluded few other studies that had no proper description of either numerator or denominator that are fundamental for representativeness and reliability of at least a period prevalence rate; no incidence could have been reported due to the lack of prospective studies on the theme, as well as ethnicity -never reported- and sequencing platform -as co-infections are diagnosed on a clinical microbiological bases by blood cultures).

  1. Line 222: "Candida spp and Clostridioides difficile", are there other fungi associated with CDI?

AR: We have reviewed once again literature to be sure on the answer and this is no; there are no other fungal co-infections, other than Candidemia, invasive candidiasis and fungal probiotics infections (mainly Saccaromyces spp, that we have discussed in lines 390-405), that have been plausibly reported as aetiologically/pathophysiologically related to Cdiff infection (indeed there are case reports describing Cdiff and other fungal infections but there is no attempt nor plausible reason or hypothesis on the fact that suffering from Cdiff could have increased the risk for the concomitant fungal infection and therefore these are off topic in respect to the topic of our review; as example Cdiff infection and pulmonary aspergillosis).

  1. Authors should talk a little about which one comes first, C. difficile? or other pathogens?

AR: we have now detailed the vicious circle on even temporal relationship in lines 117-145 and 246-277 (and depicted in Figure 1); also, more details on this temporal relationship is now provided by Table 1, where we included, when available, the time from Cdiff infection to the co-infection diagnosis and the length of the period of observation from Cdiff diagnosis to the detection of concurrent and mostly subsequent co-infections.

  1. Authors should also talk more about the intervention approach for CDI, such as antibiotics and prebiotics.

AR: we have now detailed the suggested topics in the last chapter “Practical considerations”, such as in lines 536-568 (for antibiotics and alternative treatments) and in lines 569-577 (for probiotics) and lines 577-583 (for prebiotics).

Thank you once again to Rev.2 for helping us at improving our manuscript.

Reviewer 3 Report

It is an interesting review discussing the effect of concurrent and subsequent co-infections of clostridioides difficile colitis in the era of Gut Microbiota and its impact on  Therapy. The review includes the following points : a)  Bacterial blood-stream infections and Clostridioides difficile.  b) Candida spp and Clostridioides difficile.   c)Cytomegalovirus and Clostridioides difficile.   d) . Other co-infections in Clostridioides difficile colitis.  e) Probiotics and Clostridioides difficile.       f). Fecal microbiota transplantation in Clostridioides difficile colitis.  

The review is interesting, some point to improve the quality

a) The impact of COVID19 infection on C.diffficle infection and disease prognosis

b) Since C.difficle infection is associated with colorectal cancer, it is important to expand the part of coinfection of colon cancer pathogens in case of CRC.

c) The impact of C.difficle on other GIT disease such as inflammatory bowel disease. How this could affect the therapy strategy?

Author Response

Thanks to Reviewer 3 for his/her time spent on reading our manuscript. We have reviewed a topic that is specific and refers to concurrent or subsequent infections that may be directly favored by or maybe more frequently associated (due to a pathophysiological mechanisms – mainly microbial translocation and gut mucosal immunity alterations) with clostridioides difficile infection; therefore, while extremely interesting, all the three topics suggested by the reviewer do not fit with the review target. We are sorry for not being more condescending with such suggestions, but these would lead to an extensive expansion of the manuscript with no clinical or microbiological addition to the issue of concurrent and sub-sequent co-infections in Cdiff colitis (we also didn’t discuss any casual co-infection such as meningococcal meningits and CDI, or pulmonary aspergillosis and CDI for the same reasons). Thanks anyway to Rev.3 for appreciating what we did and her/his time spent on our work.

Round 2

Reviewer 3 Report

Although the authors did not include my suggestions, the manuscript is worthy to be published.